# RULE-BASED GRID WORLD EXPLORATION UNDER UNCERTAINTY

## ABSTRACT

Grid world environments expose core challenges in sequential decision-making, including planning under partial observability and achieving sample-efficient generalization. Current Deep Reinforcement Learning methods often require millions of interactions in these structured domains, struggling to capture causal dependencies critical for efficient adaptation. We present a novel experiential learning agent with causally-informed intrinsic reward that is capable of learning sequential and causal dependencies in a robust and data-efficient way within grid world environments. After reflecting on state-of-the-art Deep Reinforcement Learning algorithms, we provide a relevant discussion of common techniques as well as our own systematic comparison within multiple grid world environments. We also investigate the conditions and mechanisms leading to data-efficient learning and analyze relevant inductive biases that our agent utilizes to effectively learn causal knowledge and to plan for rewarding future states of greatest expected return.

## 1 INTRODUCTION

Grid world environments come in many forms and have been studied extensively in the history of Artificial Intelligence, with some notable examples such as Wumpus World (Bryce, 2011), Minigrid (Chevalier-Boisvert et al., 2024), and Tileworld (Pollack & Ringuette, 1990). However, the creation of intelligent grid world agents capable of learning effectively and in a data-efficient way has posed significant challenges. Current Reinforcement Learning (RL) agents struggle in some instances due to sequential dependencies, partial observability (Wang et al., 2023a), continual learning (primacy bias Kim et al. (2024), stability-plasticity dilemma Anand & Precup (2024)), relatively high-dimensional state spaces and non-deterministic effects of actions.

Sequential dependencies usually raise the training data demand exponentially depending on the combinatorics, and, ultimately, the number of arising options, unless the agent is capable of learning causal representations that transfer well. Partial observability, on the other hand, may require the model to have access to information from prior states, which typically corresponds to the previously observed values in a grid world outside the agent's current field of view. In terms of input dimensionality, there is a trade-off between the observation window being too small for learning an effective policy and the agent's observation window being more high-dimensional, thereby demanding more training data.If the agent can represent values beyond its observation window, a learned policy must account for both the window itself and how it spatially relates to remembered information outside it.

With these considerations in mind, we introduce Non-Axiomatic Causal Explorer (NACE), a novel experiential learning agent, which leverages causal reasoning and intrinsic reward signals to enable more efficient learning as well as possesses learning mechanisms with the involved inductive biases. NACE is designed to induce causal rules from temporal and spatial local changes in the grid, it utilizes these rules to plan for and reach future states of maximum uncertainty to effectively learn more about the environment, thus improving predictability-based intrinsic reward formulations.

To illustrate the effectiveness of our approach, we provide a comprehensive discussion of state-of-the-art Deep Reinforcement Learning (DRL) techniques as well as our own systematic comparison within multiple grid worlds showing remarkable improvement in data efficiency, achieving similar performance with about 1000 samples versus DRL's requirement of 1 million samples. We examine the conditions under which learning in grid worlds attains greater data efficiency, with particular

emphasis on inductive biases facilitating close-to-optimal learning speeds without predefined inter-action rules but having the inductive biases to build them. Lastly, we analyze useful inductive biases that generalize across diverse grid worlds to enhance data efficiency in learning.

Therefore, our main contributions are:

- We propose NACE, a learning agent that induces causal rules from experience and uses them to plan and explore.
- We demonstrate that incorporating causal inductive biases can improve sample efficiency.
- We show that NACE achieves up to a 1000-fold reduction in environment interactions against a range of baselines.
- We analyze NACE's limitations and outline pathways for scaling to complex environments.

## 2 RELATED WORK

Current RL techniques, like value-based (DQN (Mnih et al., 2013)) and policy gradient-based (PPO (Schulman et al., 2017)), require millions of training steps to succeed in grid-worlds (Zhang et al., 2020), struggling to capture causal dependencies necessary for efficient planning and transferability.

Exploration improvements, such as intrinsic rewards based on information gain (Zhao et al., 2023), prediction errors (Burda et al., 2018), or visitation counts (Zheng et al., 2021; Wang et al., 2023b), enhance sample efficiency but lack structured reasoning for generalization. In contrast, Tsividis et al. (2021) propose Theory-Based RL, exemplified by EMPA, which integrates Bayesian causal mod-eling, structured exploration, and heuristic planning to generalize efficiently with minimal training. Similarly, GALOIS (Cao et al., 2022) addresses generalization by synthesizing interpretable, hier-archical programs with strict cause-effect logic, though its reliance on predefined program sketches limits flexibility in semi-structured environments. As a model-based RL, DreamerV3 (Hafner et al., 2023) learns latent state dynamics and improves behavior through imagination, enabling generaliza-tion across diverse tasks with little domain-specific adjustments. However, its reliance on learning latent representations and their dynamics reduces sample efficiency compared to methods that as-sume predefined representations.

Symbolic approaches such as STRIPS or Behavior Trees (BTs) (Guo et al., 2023; Colledanchise & Ögren, 2018) handle human-defined causal knowledge but lack lacking adaptive learning. POMDPs (Spaan, 2012) emphasize probability updates over causal discovery, while causal networks (Pearl, 1995) and structure-learning methods (Zheng et al., 2018) struggle with ambiguity and scalability.

DRL often struggles with sample efficiency, requiring substantial interactions with environments. This paper examines foundational algorithms, scalable architectures, and exploration-focused meth-ods that address these challenges.

Foundational methods include DQN Mnih et al. (2015), which combines deep neural networks with Q-learning to handle large state spaces but struggles with sparse rewards; A2C Mnih et al. (2016), which reduces variance in updates through synchronized parallel actors but is limited by its on-policy nature; TRPO Schulman et al. (2015), which ensures stable policy updates with trust region constraints but is computationally intensive; and PPO Schulman et al. (2017), which refines TRPO with clipped objectives for improved data utilization and computational efficiency.

Scalable architectures such as IMPALA Espeholt et al. (2018) address multi-task learning by lever-aging distributed architectures with off-policy corrections, offering scalability but facing synchro-nization challenges, whereby exploration-focused methods aim to address sparse rewards and com-plex state spaces. COUNT Bellemare et al. (2016) uses pseudo-counts for better exploration but is computationally demanding in large state spaces. RND Burda et al. (2018) stimulates novelty through prediction errors. However, it depends on high-quality state representations. CURIOSITY Pathak et al. (2017) rewards prediction errors of action outcomes, promoting intrinsic motivation, while RIDE Raileanu & Rocktäschel (2020) focuses on impactful actions but may struggle with ambiguous state changes. AMIGO Campero et al. (2021) generates adversarial goals to guide ex-ploration, requiring effective goal-generation mechanisms.

Model-based RL techniques can learn environment models and improve behavior through imagined future scenarios Sekar et al. (2020). DreamerV3 Hafner et al. (2023) follows this principle, using

latent state dynamics for broad applicability with minimal tuning. However, the added complexity of learning latent space representations along with their dynamics leads to less sample efficiency compared to when representations are already present, leaving only the dynamics to be learned.

# 3 NON-AXIOMATIC CAUSAL EXPLORER

**NACE** is our proposed experiential learning technique with causality-informed intrinsic reward and strong inductive biases for grid worlds to boost sample efficiency. Here, we provide formal descriptions of NACE. For a comprehensive list of symbols see Appendix C.

## 3.1 STATES AND RULE REPRESENTATION

State in NACE is a tuple $s = (s_{\text{spatial}}, s_{\text{internal}})$ consisting of a two-dimensional value array $s_{\text{spatial}} \in \mathbb{N}^{m \times n}$ and a one-dimensional value array $s_{\text{internal}} \in \mathbb{N}^k$, as shown in Figure 1. The two-dimensional array reflects the spatial structure in the grid world, including remembered cells beyond the current view, while the one-dimensional array is used for internal values, such as inventory items.

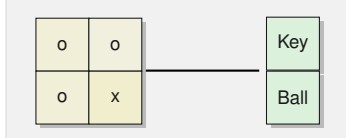

Figure 1: State components

Learned rules have the form of $(preconditions, action) \Rightarrow consequence$ where the precondition holds a conjunction of cell value constraints spatially relative to the cell value of the consequence, and the consequence predicts one particular cell's value as well as the values of the one-dimensional array at the next timestep as depicted in Figure 2.

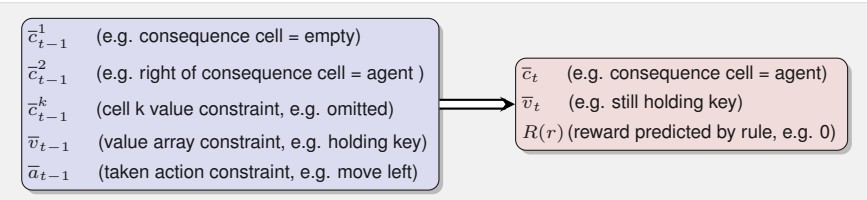

Figure 2: Rule schema: preconditions with an action and consequence

Each rule tracks evidence using counters for $w_+$ and $w_-$, similar to (Wang, 2013), which measure the accuracy of the rule's predictions. Positive evidence ($w_+$) is accumulated whenever a perfectly matching rule predicts correctly, while negative evidence ($w_-$) increases with incorrect predictions. Tracking of evidence helps the agent refine its causal knowledge by prioritizing more reliable rules. Examples of created rules are provided in Appendix G.

## 3.2 INDUCTIVE BIASES

It is well-known that favorable inductive biases can enhance sample efficiency. Below are inductive biases that are incorporated in NACE and relevant for grid world environments:

1. **Temporal Locality**: NACE constructs rules based solely on the current and previous state, modeling relevant dependencies locally in time.

2. **Causal Representation**: NACE's knowledge representation is centered around the causal rules, which can be chained and are independent of the objective.

3. **Spatial Equivariance**: Ability to model causal dependencies between grid cells independently of the specific location of the cells considered in the dependency, meaning learned rules are applied at any location.

4. **State Tracking**: Ability to effectively track states outside of the field of view of the agent based on the recorded or estimated locations. NACE explicitly keeps track of a bird's-eye view map by recording observations and updating the values that are within its observability window.

5. **Attentional Bias**: Relevant dependencies tend to involve values that have either observably changed or a different value than predicted. Only rules that show a change from the previous to the current timestep, or differ from the predicted value, are considered for rule formation, evidence updating, and prediction.

Additional discussions on inductive biases as well as ablation studies can be found in Appendix B.

### 3.3 CURIOSITY MODEL

Curiosity model outlines the mechanism that helps NACE systematically acquire missing causal knowledge about the environment. The key principle is realized by making the agent plan to reach a state that it is most unfamiliar with. The familiarity is judged by whether existing rules match well with the situation, whereby matching is a matter of degree dependent on how many rule conditions match the cells in the known state. This motivates the following formalism:

- *Match* value of a rule $r$ is evaluated relative to consequence cell $c$:

$$M(r, c) = \frac{\text{Number of matched preconditions}}{\text{Total number of preconditions}}$$

- *Cell match* value of a cell $c$ dependent on all $m$ existing rule match values:

$$C(c) = \max(0, M(r_1, c), \dots, M(r_m, c))$$

- *State match* value $S(s)$ of a state $s$ is the maximum $C(c)$ value of its cells, whereby a value smaller than 1 indicates uncertainty from imperfect rule matches. This value is the secondary "explorative" objective in the planning process that guides the agent's decisions:

$$S(s) = max(C(c_1), \dots, C(c_m))$$

### 3.4 NACE ARCHITECTURE

Figure 3 illustrates the high-level architecture of NACE, which consists of several interconnected components working together to enable learning and decision-making. The related pseudocode is provided in Appendix D.

1. **Observer:** Its role is to update a bird's-eye view map via values from the partial observation 2D array, then to find changes in input, as well as prediction-observation mismatches (prediction failures). Formally, this corresponds to determining the sets:

    - **Set of changes in observations**: $M_t^{\text{change}} = \{c_{t,x,y}^{\text{observation}} \mid c_{t,x,y}^{\text{observation}} \neq c_{t-1,x,y}^{\text{observation}}\}$
      This set captures all grid cells $c_{x,y}$ where the observed value has changed between timesteps $t-1$ and $t$, highlighting areas that have been updated or modified.
    - **Set of observation mismatches**: $M_{\text{mismatched},t}^{\text{observation}} = \{c_{t,x,y}^{\text{observation}} \mid c_{t,x,y}^{\text{prediction}} \neq c_{t,x,y}^{\text{observation}}\}$
      This set includes all grid cells where the observed value differs from the predicted value at time $t$, indicating potential prediction failures.
    - **Set of prediction mismatches**: $M_{\text{mismatched},t}^{\text{prediction}} = \{c_{t,x,y}^{\text{prediction}} \mid c_{t,x,y}^{\text{prediction}} \neq c_{t,x,y}^{\text{observation}}\}$
      This set identifies all grid cells where the predicted value does not match the observed value at time $t$, from the perspective of predictions.

    These sets enable the Observer to track state changes and prediction failures, ensuring an accurate understanding of the environment and supporting the system's adaptive and predictive capabilities.

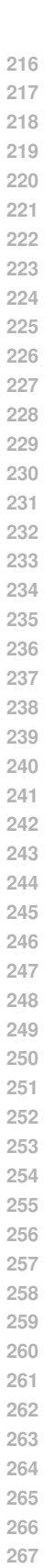
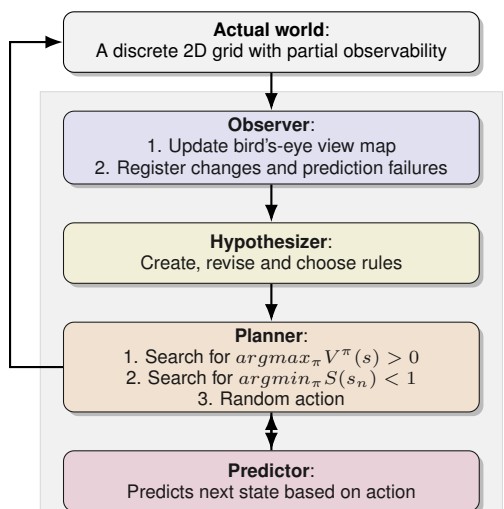

**Actual world**:
A discrete 2D grid with partial observability

**Observer**:
1. Update bird's-eye view map
2. Register changes and prediction failures

**Hypothesizer**:
Create, revise and choose rules

**Planner**:
1. Search for $argmax_\pi V^\pi(s) > 0$
2. Search for $argmin_\pi S(s_n) < 1$
3. Random action

**Predictor**:
Predicts next state based on action

**Actual world** represents the real simulated 2D grid environment (Minigrid) with a cell-granular partial observability model. In each frame, the field-of-view local to the agent is passed on to the observer.

**Observer** takes the field-of-view 2D array as input and detects changes in values as well as identifies prediction failures from rules that predict incorrectly.

**Hypothesizer** creates and updates rules based on whether their predictions align with observations, whereby only changed-cells and prediction-mismatch cells as reported by Observer are considered.

**Planner** searches for optimal actions that lead to greater-than-zero expected return, and if none such is found, searches for actions that lead to a state of lowest state match value greater than zero. Finally, in case such also does not exist, a random action is chosen

**Predictor** forecasts the next state from the current state and the taken action, utilizing individual rules to predict a state transition of the entire state, whereby for each cell its predicted value comes from the rule with the highest $M(r, c)$.

Figure 3: Flow diagram of the system

2. **Hypothesizer:** Associating positive and negative evidence based on prediction success, as well as creating new rules when positive evidence is initially found.

   Formally, for each rule $r = ((\overline{c}_{t-1}^1 \wedge ... \wedge \overline{c}_{t-1}^k \wedge \overline{v}_{t-1} \wedge \overline{a}_{t-1}) \Rightarrow (\overline{c}_t \wedge \overline{v}_t \wedge R(r)))$,
   $\overline{c} := (c_r = c)$ indicates that the value in the rule precondition aligns with the actual cell value, value array in case of $\overline{v}$, and taken action in case of $\overline{a}$.

   The rule preconditions are met when all *equality constraints* $\overline{c}_{t-1}^1, ..., \overline{c}_{t-1}^k, \overline{v}_{t-1}, \overline{a}_{t-1}$ hold. Positive evidence is attributed when the equality constraints of the postcondition $\overline{c}_t, \overline{v}_t$ are met as well and the predicted reward aligns with the observed reward ($R_t = R(r)$). To increase computational efficiency, only cells that changed value or have a different value than predicted are considered. Formally, this is being defined as:

   $$w_+(r) = \begin{cases} w_+(r) + 1 & \text{if } \{c_{t-1}^1, ..., c_{t-1}^k, c_t\} \subseteq M \\ w_+(r) & \text{otherwise} \end{cases}$$

   $$\text{where } M = (M_t^{change} \cup M_{\text{mismatched}_t}^{\text{observation}})$$

   Negative evidence is assigned when any of the postcondition equality constraints are unmet:

   $$w_-(r) = \begin{cases} w_-(r) + 1 & \text{if } c_t \in M_{\text{mismatched}_t}^{\text{prediction}} \\ w_-(r) & \text{otherwise} \end{cases}$$

   Finally, rules $r$ for which $w_-(r) > w_+(r)$ become inactive, and for two rules $r_1, r_2$ if their preconditions match (including the action) but the postconditions are different, only the rule with the higher truth expectation $T_{exp}(r)$ is selected, which is calculated according to:

   $$w(r) = w_-(r) + w_+(r), \quad frequency(r) = \frac{w_+(r)}{w(r)}, \quad confidence(r) = \frac{w(r)}{w(r)+1}$$
   $$T_{exp}(r) = (frequency(r) - \tfrac{1}{2}) * confidence(r) + \tfrac{1}{2}$$

   This not only allows the system to find the relevant preconditions under which a consequence happens when the action is utilized, but also gives the system tolerance to non-deterministic effects and enables accounting for uncertainty. More details in Appendix A.

3. **Planner**: NACE makes use of a depth- and width-bounded Breadth-First-Search algorithm with a combined search objective consisting of two components: it searches for states resulting from the different action sequences for futures that lead to the max. Expected return or, if not existing, the lowest state match value. Hence, it applies a key RL principle to maximize the expected long-term return (Sutton et al., 1999), with the policy determined by the considered action sequence: $\pi(t) = a_t$ for $t = 1, 2, \ldots, n$ whereby $n$ is smaller-or-equal (dependent on where the optimum is found) to the maximum planning horizon:

$$\pi(t) = \begin{cases} \arg\max_\pi V^\pi(s_0) & \text{if } V^\pi(s_0) = R_{exp} \\ \arg\min_\pi S(s_n) < 1 & \text{otherwise} \end{cases}$$

$$\text{where } R_{exp} = \mathbb{E}\left[\sum_{t=0}^n \gamma^t R(s_t) \mid s_0 = s, \pi\right] > 0$$

According to this definition, if no return greater than zero can be obtained for any considered action sequence, the system instead plans for a future state of lowest state match value, while maintaining the "uncertainty-at-the-last-step" condition:

$$(\forall t : (0 \le t < n) \rightarrow S(s_t) = 1) \wedge S(s_n) < 1$$

meaning the action sequence is constrained to be planned in such a way that the state match value is 1 except for the last action, where it is minimized for the resulting state. Such constraint maximizes the agent's chance to reach the state of minimum state match while ensuring this low match value is not a consequence of predicting from states where the knowledge was not fully utilized.

Due to the number of possible options, the planning algorithm dominates the asymptotics of NACE. It has the computational complexity of $O(|V| + |E|)$ where $V$ and $E$ are the sets of nodes and edges of the search graph. Constant-bounded search depth and width can be achieved by pruning of branches by expected return and state match value, however, bounded search depth can negatively affect performance, as analyzed in Appendix A.

4. **Predictor**: When the planner queries for the predicted state from a given state and an action, the role of the predictor is to construct the predicted state by applying all knowledge to the given state in the following way: initializing with the cell values from the given state, where for each cell we utilize only the rule $r$ with $M(r, c) = 1$ and maximum $T_{exp}(r)$, meaning the rule preconditions match perfectly to the given state, the action that has been considered, and $r$ has the highest truth expectation among the rule candidates.

In this case, the postcondition cell value of the rule is applied to the corresponding cell at position $(x, y)$ in the predicted state, while else the cell keeps the value from the previous state. Hence, for utilized rules $r^* = ((\bar{c}_t^1 \wedge ... \wedge \bar{c}_t^k \wedge \bar{v}_t \wedge \bar{a}_t) \Rightarrow (\bar{c}_{t+1} \wedge \bar{v}_{t+1} \wedge R(r^*)))$, where $\bar{c}_{t+1}$ and $\bar{v}_{t+1}$ constrains the cell value and value array of the consequence:

$$c_{t+1,x,y} = \begin{cases} c_{t+1} & \text{if } r^* = \underset{r|M(r,c_{t,x,y})=1}{\arg\max} T_{exp}(r) \\ c_{t,x,y} & \text{otherwise} \end{cases}$$

Now, while $s_{t+1}$ is a composition of the cells at all locations at time $t + 1$, the reward associated with $s_{t+1}$ is the average reward of each of the $N$ utilized rules:

$$R(s_{t+1}) = \frac{1}{N} \sum_{i=1}^N R(r_i^*)$$

## 4 EXPERIMENTS IN MINIGRID

To evaluate the effectiveness of NACE compared to other DRL techniques, we conducted a series of experiments in Minigrid Chevalier-Boisvert et al. (2024), a 2D grid world environment featuring diverse and procedurally generated scenarios (Hardware setup including test environments are found in Appendices F and H). We focus on Minigrid levels that feature partial observability, challenging the agent to operate with limited information about its surroundings. The selected environments are categorized based on their specific characteristics as **static** (fixed start & goal locations), **dynamic** (random positions of start, goal, and obstacle), and **dynamic with sequential dependencies** (tasks requiring specific action sequences such as a door that needs a key).

In each environment, we recorded the average reward, episode length, and standard deviation every 100 timesteps, whereby each timestep incorporates the observed state, action taken, and obtained reward. Below we present results for each category, using the selected RL techniques from Section 2 with Behavior Trees (BTs) and hard-coded policies employed as performance upper bounds.

**Hyperparameter Choices:** NACE used a planning horizon of 100 steps and a truth expectation threshold of 0.5 for rule filtering. For DRL baselines, we used Stable Baselines3 (for PPO, TRPO, A2C, DQN) and Torchbeast (for IMPALA and intrinsic reward methods), all configured for consistent observation processing. Baselines shared a 4-layer CNN with $2 \times 2$ kernels, 16–128 filters, and

ReLU activations. PPO employed a learning rate of $3\times10^{-4}$, batch size 64, 10 epochs per update, and 2048 rollout steps. TRPO adopted a learning rate of $10^{-3}$ and a KL-divergence constraint 0.01. A2C ran with five rollout steps and a learning rate of $7\times10^{-4}$. DQN operated with a buffer size of 1M, soft target update every 10k steps, and learning rate $10^{-4}$. Exploration-based methods (e.g., RIDE, RND, AMIGO) used a 3-layer CNN (3×3 kernels), an LSTM, and an intrinsic reward coefficient of 0.1. COUNT relied on 128-bit pseudo-count hashes and reset probability 0.001. DreamerV3 followed the official implementation with default settings, matching the action repeat and frame stack to our environment. Similarly, the BTs were configured with access to the true shortest path, serving as upper-bound reference policies. All agents applied a discount factor $\gamma = 0.99$, gradient clipping, and default settings from their respective frameworks. Additional details are found in Appendix E.

## 4.1 STATIONARY ENVIRONMENTS

In this category, because the start and goal locations are fixed, the primary challenge for the agent is to consistently learn and optimize navigation strategies over repeated episodes.

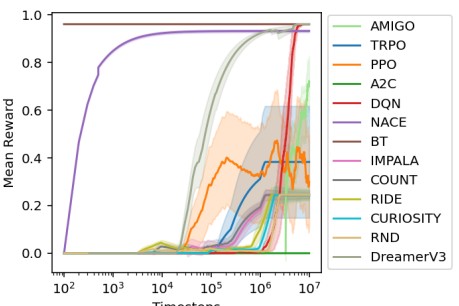

| Techn. | Avg. reward | S. dev. |
|---|---|---|
| TRPO | 0.383 | 0.469 |
| PPO | 0.763 | 0.381 |
| A2C | 0.000 | 0.000 |
| DQN | **0.961** | **0.000** |
| IMPALA | 0.245 | 0.027 |
| COUNT | 0.243 | 0.025 |
| RIDE | 0.245 | 0.036 |
| CURIOSITY | 0.245 | 0.041 |
| RND | 0.245 | 0.049 |
| AMIGO | 0.778 | 0.203 |
| DreamerV3 | **0.961** | **0.000** |
| NACE | 0.932 | 0.011 |

Figure 4: Learning curves and performance metrics for the *MiniGrid-DistShift2-v0*

**MiniGrid-DistShift2-v0:** In this environment the fixed start and goal locations are accompanied by stationary lava obstacles, which the agent must navigate around to reach the goal. DQN and DreamerV3 perform well, achieving a near-optimal policy with an average reward of 0.96, closely mirroring the performance of the BT. NACE reached a lower value of 0.93, while it was three orders of magnitude more sample-efficient. The next-best policies were found by AMIGO and PPO with an average reward of 0.78 and 0.76, while the others were below 0.5 and less sample-efficient.

## 4.2 DYNAMIC ENVIRONMENTS

Given that the start and goal locations, along with obstacle positions, are randomized in each episode, these environments require the agent to continuously adapt to new and unpredictable conditions.

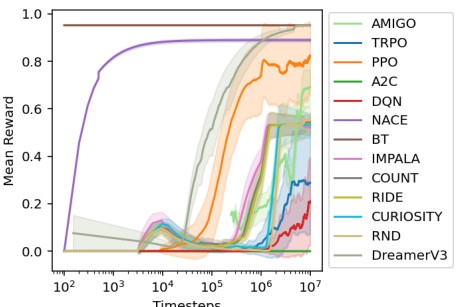

| Techn. | Avg. reward | S. dev. |
|---|---|---|
| TRPO | 0.187 | 0.375 |
| PPO | 0.838 | 0.309 |
| A2C | 0.000 | 0.000 |
| DQN | 0.114 | 0.309 |
| IMPALA | 0.521 | 0.064 |
| COUNT | 0.543 | 0.067 |
| RIDE | 0.535 | 0.070 |
| CURIOSITY | 0.531 | 0.103 |
| RND | 0.551 | 0.111 |
| AMIGO | 0.690 | 0.151 |
| DreamerV3 | **0.952** | **0.005** |
| NACE | 0.922 | 0.013 |

Figure 5: Learning curves and performance metrics for *MiniGrid-LavaGapS7-v0*

**MiniGrid-LavaGapS7-v0:** Here, the agent must navigate around randomly placed lava obstacles to reach a fixed goal, requiring adaptability due to the varying paths between episodes. The 5x5 free space - mostly covered by the agent's observation window - is complicated by dynamically spawning lava. From Figure 5, DreamerV3 emerges as the most effective, closely followed by NACE, whereby

NACE takes about $10^3$ timesteps compared to DreamerV3 taking $3 \times 10^5$ to reach a mean reward of around $0.8$. BT's optimal policies are similar in performance to DreamerV3, while PPO (reaching $0.838$) shows instability in learning and greater sensitivity to initialization, as indicated by a higher standard deviation. Others performed poorly, despite the level is nearly fully observable.

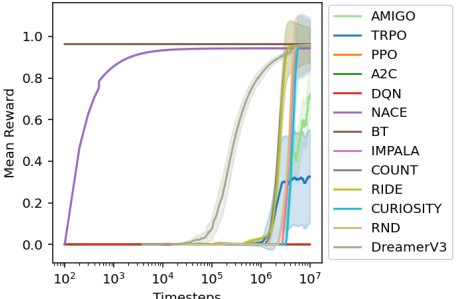

| Techn. | Avg. reward | S. dev. |
|---|---|---|
| TRPO | 0.381 | 0.467 |
| PPO | 0.000 | 0.000 |
| A2C | 0.000 | 0.000 |
| DQN | 0.000 | 0.000 |
| IMPALA | 0.958 | 0.238 |
| COUNT | **0.960** | **0.168** |
| RIDE | 0.959 | 0.170 |
| CURIOSITY | 0.958 | 0.261 |
| RND | 0.958 | 0.222 |
| AMIGO | 0.778 | 0.203 |
| DreamerV3 | 0.954 | 0.008 |
| NACE | 0.943 | 0.005 |

Figure 6: Learning curves and performance metrics for *MiniGrid-SimpleCrossingS11N5-v0*

**MiniGrid-SimpleCrossingS11N5-v0:** Here the agent faces a large grid with multiple intersections and potential dead ends. The randomized layout in each episode forces the agent to develop a robust exploration strategy. As Figure 6 shows, DreamerV3, IMPALA, COUNT, RIDE, CURIOSITY, RND and NACE achieved near-optimal policies since their intrinsic reward mechanisms seem to be particularly helpful in the environments where the observable window covers only a small part. AMIGO found reasonable policies with a reward of $0.78$, while the others scored below $0.5$.

### 4.3 Dynamic Environments with Sequential Dependencies

In these environments, the need to perform actions in a specific sequence adds complexity and tests the agent's ability to plan and execute multi-step strategies.

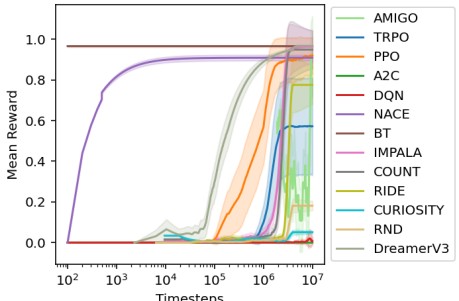

| Techn. | Avg. reward | S. dev. |
|---|---|---|
| TRPO | 0.577 | 0.471 |
| PPO | 0.890 | 0.263 |
| A2C | 0.000 | 0.000 |
| DQN | 0.000 | 0.000 |
| IMPALA | 0.964 | 0.162 |
| COUNT | 0.949 | 0.185 |
| RIDE | 0.775 | 0.188 |
| CURIOSITY | 0.051 | 0.016 |
| RND | 0.181 | 0.046 |
| AMIGO | 0.932 | 0.388 |
| DreamerV3 | **0.967** | **0.003** |
| NACE | 0.909 | 0.028 |

Figure 7: Learning curves and performance metrics for *MiniGrid-Unlock-v0*

**MiniGrid-Unlock-v0:** In this scenario, the agent must first locate and pick up a key before unlocking a door to reach the goal and obtain the reward. This sequential dependency adds a layer of complexity that challenges the agent's ability to plan ahead. Even though it is a single sequential dependency, the DRL techniques that learned the fastest initially, DreamerV3 and PPO, demands almost a million timesteps to converge to a similarly effective policy as NACE, which achieves this within just $10^3$ steps (Figure 7). While PPO scored $0.89$ and showed more instability in learning, it is far less chaotic than AMIGO. IMPALA reached the optimal policy after about 2 million steps, performing similarly well to COUNT and AMIGO in the end. Also, in our runs, TRPO did not exceed a mean episode reward of $0.6$, while A2C and DQN failed to learn any effective policy.

**MiniGrid-DoorKey-8x8-v0:** This environment introduces an additional layer of sequential dependency by requiring the agent to navigate through an unlocked door to reach a goal in a separate room. While passing through the door adds complexity, the primary challenge lies in the sparse reward structure, as no reward is given for merely using the door, since only reaching the final goal is rewarded. DreamerV3 and COUNT nearly achieved the optimal policy with a reward of $0.975$ and $0.96$ (Figure 8). AMIGO reached $0.87$ within $10^7$ timesteps, which is below the average reward

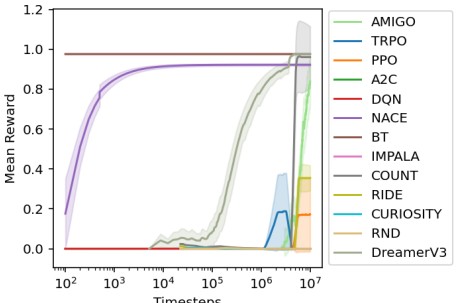

| Techn. | Avg. reward | S. dev. |
|---|---|---|
| TRPO | 0.000 | 0.000 |
| PPO | 0.156 | 0.357 |
| A2C | 0.000 | 0.000 |
| DQN | 0.000 | 0.000 |
| IMPALA | 0.000 | 0.000 |
| COUNT | 0.960 | 0.308 |
| RIDE | 0.354 | 0.126 |
| CURIOSITY | 0.000 | 0.000 |
| RND | 0.000 | 0.001 |
| AMIGO | 0.868 | 0.241 |
| DreamerV3 | **0.977** | **0.004** |
| NACE | 0.922 | 0.012 |

Figure 8: Learning curves and performance metrics for *MiniGrid-DoorKey-8x8-v0*

of NACE requiring only $10^4$ steps. Overall, the results suggest poor combinatorial scaling of the involved DRL techniques, while NACE needed a similar number of steps as for *MiniGrid-Unlock-v0* to learn an effective policy.

## 5 DISCUSSION

**Sample Efficiency and Generality:** The observed sample efficiency of NACE originates from explicitly exploiting the cell-based grid world state observations to create transition rules. While the inductive biases are favourable for grid world environments, they make NACE less generic than DreamerV3 and demand the application of various feature extraction techniques (variants of Semantic Simultaneous Localization and Mapping Qi et al. (2020), to generate semantic gridmaps) to be applied outside of the grid world environments. However, DreamerV3 is not designed to be directly applicable to real-world applications either, but has been showcased exclusively in game-like scenarios with a sample efficiency insufficient for real-time learning outside of simulation. Additional discussions, such as about NACE's near-optimal performance characteristics and representational limitations, can be found in Appendix A.

**Inductive Bias Ablation Study:** We evaluated the contribution of NACE's inductive biases through targeted ablations, detailed in Appendix B. Removing *temporal locality* led to an exponential growth in rule candidates, severely degrading performance. Without the *attentional bias*, which restricts rule creation to changed or mismatched cells, rule learning became inefficient, overfitting to irrelevant input. Eliminating *spatial equivariance* prevented generalization across grid locations, reducing sample efficiency by more than an order of magnitude. Disabling *state tracking* caused the agent to loop or revisit known areas, lacking the memory needed to navigate efficiently. Lastly, the *causal representation*, NACE's condition-action-effect rule formalism, defines its core reasoning and is not ablatable. Together, these results show that each bias is essential for tractable and generalizable causal learning, and further analysis of the contribution of each can be found in Appendix B.

## 6 CONCLUSION

We introduced NACE, an experiential learning agent designed to enhance data efficiency in grid world environments by leveraging causally-informed intrinsic rewards and strong inductive biases. We compared NACE with state-of-the-art DRL techniques, demonstrating that while these techniques are able to eventually achieve near-optimal policies, they often require significantly more data, especially as task complexity increases due to factors such as sequential dependencies. NACE, by contrast, extends the RL framework to empirically support causal relations, enabling effective learning and decision-making even in low-data settings without relying on pre-defined causal models. Our causality-informed curiosity model, combined with the outlined inductive biases, facilitates systematic exploration and learning requiring significantly fewer timesteps. We hope that future work in the field will strike new compromises regarding the inclusion of inductive biases, leading to highly sample-efficient DRL that retains the ability to converge to optimal policies. Moving forward, we plan to generalize NACE to handle three-dimensional and continuous spaces, as well as explore neural implementations of NACE, further advancing the capabilities of learning agents.

## REPRODUCIBILITY STATEMENT

- We utilized open-source implementations of the selected DRL algorithms from public repositories (not including our technique):
  - AMIGO was from here: `https://github.com/facebookresearch/adversarially-motivated-intrinsic-goals`
  - BT is here: `https://github.com/andreneco/minigrid_bt`
  - DQN, A2C, TRPO, and PPO were established on Stable Baselines3 (SB3)'s baselines repository (Raffin et al., 2021): `https://stable-baselines3.readthedocs.io/`
  - DreamerV3 was from here: https://github.com/qxcv/dreamerv3
  - All the other were from here: `https://github.com/sparisi/cbet`
- We used the the Minigrid package for the environments in our comparison, which is available here: `https://github.com/Farama-Foundation/Minigrid`
- For NACE we provide a stand-alone zip archive for reviewers to reproduce our results, which is runnable on a regular computer with Python interpreter. It includes a README.txt in the NACE folder, as well as scripts to generate the tables and the plots present in the paper.

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

## APPENDIX A  ADDITIONAL STUDIES AND DISCUSSIONS

To analyze the factors that lead NACE to finding close-to-optimal policies, as well as factors which can, in some circumstances, hamper it from learning such, we present contributing factors from a conceptual perspective, examine the impact of hyperparameter choices, and assess robustness to non-determinism arising from random action consequences.

- **Representational Limitations**: NACE's rule-based framework captures only spatially relative dependencies from one timestep to the next. It does not exploit the inherent structural statistics of environment generation, which are leveraged by various DRL techniques. While these structural dependencies are most apparent in static environments where locations remain constant, they are also present in dynamic environments. For example, the goal location consistently appears in the bottom-right corner not only in *MiniGrid-Empty* levels but also in *MiniGrid-SimpleCrossingS11N5-v0*. NACE's inability to utilize these broader environmental patterns could limit its performance compared to methods that can, even though in Minigrid, this issue was less prevalent.

  Additionally, NACE's rules are tied to an action, meaning agent-external changes that are not caused by NACE need to be learned for each action separately, considerably lowering its sample efficiency by a factor of the number of actions. The mechanism could be extended to incorporate learning of rules without an action as a precondition, leaving it to evidence collection whether the action is considered dependent on the truth expectation of the alternative rules.

- **Study of Reduced Planning Horizon**: NACE's estimation of expected returns relies heavily on its planning horizon. Short planning horizons can significantly reduce performance, especially in tasks requiring long-term planning. To quantify this effect, we examine two cases: *MiniGrid-DoorKey-8x8-v0*, which demands longer-horizon planning, and *MiniGrid-DoorKey-6x6-v0*, which is less demanding in this regard. As shown in Table 1, running NACE with a planning horizon of only eight steps in *MiniGrid-DoorKey-8x8-v0* results in convergence to an average return of 0.48, whereas extending the horizon to 100 steps improves the average reward to 0.92. In contrast, in *MiniGrid-DoorKey-6x6-v0*, NACE maintains an average reward of 0.93 regardless of the planning horizon. A similar pattern is observed in *MiniGrid-Empty-16x16-v0*, where the average reward drops from 0.91 to 0.41 when the planning horizon is reduced from 100 to 8 steps. These results highlight NACE's dependence on adequate planning horizons for effective rule chaining and the significant performance degradation that occurs when the planning horizon is too short.

| Environment | Planning Horizon, Average Reward |
|---|---|
| MiniGrid-DoorKey-6x6-v0 | 8 steps, 0.93 |
| MiniGrid-DoorKey-6x6-v0 | 100 steps, 0.93 |
| MiniGrid-DoorKey-8x8-v0 | 8 steps, 0.48 |
| MiniGrid-DoorKey-8x8-v0 | 100 steps, 0.92 |
| MiniGrid-Empty-16x16-v0 | 8 steps, 0.41 |
| MiniGrid-Empty-16x16-v0 | 100 steps, 0.91 |

Table 1: NACE Performance with different planning horizons

- **Robustness to Non-Determinism**: NACE's rule representation incorporates uncertainty handling through evidence counters, enabling it to cope with non-deterministic state transitions. To assess this capability, we modify the environment to invoke unintended actions with certain probabilities. In *MiniGrid-Empty-16x16-v0*, when 10% of actions result in unintended outcomes, NACE still achieves an average reward above 0.9, demonstrating basic tolerance to non-determinism. However, when the probability of unintended actions increases to 20%, NACE fails to complete the task within the maximum allowed time in all episodes. Higher tolerance to non-determinism can be achieved by increasing the default truth expectation threshold for rule usage above the default value of 0.5. However, this adjustment reduces sample efficiency, as it requires the agent to confirm each rule multiple times before utilizing it.

## APPENDIX B    ABLATION STUDY: EFFECTS OF OMITTING KEY INDUCTIVE BIASES IN NACE, AND INDUCTIVE BIASES IN DRL

### B.1    ESSENTIAL INDUCTIVE BIASES IN NACE

To ensure tractability and generalization, NACE incorporates five key inductive biases, each critical for data-efficient causal learning discussed below.

| Inductive Bias | Role in NACE | Failure Mode if Omitted |
|---|---|---|
| **Temporal Locality** | Rules only from consecutive timesteps | Rule explosion from considering longer histories |
| **Attentional Bias** | Considers only changed or mismatched cells | Inefficient rule learning over unchanged areas |
| **Spatial Equivariance** | Applies rules at any grid location | Must relearn rules per location ($\times$ grid size) |
| **State Tracking** | Maintains memory of previously seen tiles | Agent loops or re-explores known areas |
| **Causal Representation** | Models condition-action-effect rules | Not ablatable, foundational to NACE |

Table 2: Essential inductive biases in NACE and consequences of omitting them.

### B.2    CAUSAL RULE REPRESENTATION

The causal rule representation is foundational to NACE's operation and cannot be omitted. However, we analyze the effects of reducing the planning horizon, which limits the depth of chaining, in Appendix A.

### B.3    TEMPORAL LOCALITY AND ATTENTIONAL BIAS

Omitting these biases with larger environment sizes is infeasible due to the combinatorial explosion of potential rules, as we will now analyze.

- For an environment of size $w \times h$, the number of possible rule preconditions for a single timestep is $2^{w \cdot h}$, as each particular cell can either be considered or not be considered in the precondition of a rule.

- For a time window of duration $d$, this expands to $2^{w \cdot h \cdot d}$, leading to 18446744073709551616 possible rule preconditions for an $8 \times 8$ grid within a single timestep.
- NACE is tied to the Markov Assumption, particularly within the observational window, as all rule construction and updating consider only the previous and current state. However, its bird-view map representation also contains values from observations of previous time steps, which are currently out of view of the agent, which brings us to the next point, state tracking.

## B.4 STATE TRACKING

Without state tracking, NACE lacks memory of prior observations and memory of observations that are outside of its field of view. This results in oscillatory behavior caused by the exploration strategy of the agent, as it can only utilize the visible information.

- In our experiments across 10 runs in MiniGrid-Empty-8x8-v0, this led the agent to turn indefinitely due to the curiosity model assigning low match values to previously visited areas (due to the lack of state tracking they are always considered to be of unknown value) which are now outside of the field-of-view of the agent. The closest such cell is immediately behind the agent with the default partial observation model in Minigrid, which explains the behavior.
- State Tracking plays a critical role in ensuring purposeful exploration and decision-making, for the agent to know which places have been visited and what it has been observing at the particular locations, as well as which locations have yet to be observed.
- Sequential dependencies often depend on state tracking. An example of this is when a door has to be opened with a key, where the key and the door are too far apart to be observed concurrently, demanding some form of spatial memory. Another form of state tracking lies in the observable inventory array, which, when absent, would need the modeling of long-range temporal dependencies (e.g., did the agent already pick up the key?), which would demand a suitable model structure to be learnable by the agent.

## B.5 SPATIAL EQUIVARIANCE

The absence of spatial equivariance significantly impacts sample efficiency.

- Each rule must be learned independently for every location, meaning in an 8x8 grid, the agent has to learn 64 times the same set of rules. However, since particular arrangements of cell values will not re-appear through the environment generation, it can take significantly longer to learn the relevant knowledge without this bias.
- Hence for the general case with an environment of size $w \times h$, this increases the required sample count at least by a factor of $w \cdot h$, harming significantly the sample efficiency of the technique.
- Conceptually, we also would like to point out that the rule learning mechanisms do not allow to learn spatial equivariance retrospectively either, while some DRL techniques, dependent on the model structure, could potentially acquire it.

These results highlight the necessity of each inductive bias in ensuring the scalability, efficiency, and functionality of NACE.

## B.6 WHICH INDUCTIVE BIASES ARE PRESENT IN THE DRL TECHNIQUES

In the main paper, we outlined the inductive biases of NACE. However, we would like to point out that some of them are also inherent in the DRL techniques, complementing our discussion on inductive biases in DRL and NACE:

- **Temporal Locality**: The DRL methods perform best when the Markov Assumption is met, despite LSTM allowing to cope with partial observability, the need to capture long-range temporal dependencies makes sample-efficient learning more difficult.

- **Causal Representation**: While not explicitly stated as a set of cause-effect relations, DreamerV3's learned dynamics model can predict the consequence states of actions, which is not the case for the model-free DRL methods. Such modeling is to some extent independent from the objective (what is rewarding), and allows an agent to train itself from simulated experience by predicting novel states, and to reach novel goals.

- **Spatial Equivariance**: Clearly the DRL techniques do not have an explicit rule representation, however the Convolution layers in the DRL policies allow for learned features to be identified at different locations, improving generalization.

- **State Tracking**: Is not explicitly handled by the DRL techniques as a separate point, instead it is handled in the same way as non-local temporal dependencies in the LSTM-including policies, while NACE builds a bird view map explicitly, which can be considered to be a form of spatial memory.

- **Attentional Bias**: While NACE has a strong prior for which cells to consider based on observably changed values and prediction mismatches, the DRL policies with Convolution layers are more flexible and allow an agent to learn which values are relevant in relation to each other.

## APPENDIX C    NOTATION AND SYMBOLS

| Symbol | Description |
|---|---|
| $s$ | State, represented as a combination of a 2D grid ($s_{grid}$) and a 1D array ($s_{array}$) |
| $a$ | Action taken by the agent |
| $r$ | Causal rule in the form (preconditions, action) $\Rightarrow$ consequence |
| $c_{t,x,y}$ | Cell value at position $(x, y)$ in the 2D grid at time t |
| $\bar{c}$ | Equality constraint on a cell value (e.g., $c_r = c$) |
| $v_t$ | Value array at time t |
| $\bar{v}$ | Equality constraint on value array (e.g., $v_r = v$) |
| $M(r, c)$ | Match value of a rule $r$ for cell $c$, based on the fraction of preconditions satisfied |
| $C(c)$ | Cell match value for cell $c$, derived from the maximum match value across all rules |
| $S(s)$ | State match value for state $s$, calculated as the average $C(c)$ for cells with $C(c) > 0$ |
| $w_+(r)$ | Positive evidence counter for rule $r$, incremented when predictions align with observations |
| $w_-(r)$ | Negative evidence counter for rule $r$, incremented when predictions differ from observations |
| $w(r)$ | Total evidence count for rule $r$, defined as $w(r) = w_+(r) + w_-(r)$ |
| $frequency(r)$ | Fraction of positive evidence for rule $r$, defined as $f(r) = \frac{w_+(r)}{w(r)}$ |
| $confidence(r)$ | Confidence factor for rule $r$, defined as $c(r) = \frac{w(r)}{w(r)+1}$ |
| $f_{\exp}(r)$ | Expected truth value for rule $r$, calculated as $f_{\exp}(r) = (f(r) - \frac{1}{2}) \cdot c(r) + \frac{1}{2}$ |
| $M_t^{change}$ | Set of cells with changes in observed values between timesteps $t-1$ and $t$ |
| $M_{\text{mismatched},t}^{observation}$ | Set of cells where observed values differ from predicted values at timestep $t$ |
| $M_{\text{mismatched},t}^{prediction}$ | Set of cells where predictions differ from observations at timestep $t$ |
| $R(r)$ | Reward associated with rule $r$ |
| $R(s)$ | Reward associated with a state $s$, defined as the average reward of rules applied to generate $s$ |
| $V(s)$ | Value of state $s$, used in planning for maximizing long-term returns |
| $\pi(t)$ | Planned action sequence or policy at timestep $t$ |
| $\gamma$ | Discount factor for future rewards |

## APPENDIX D    PSEUDOCODE

The system can be described by the pseudocode:

---

### Algorithm 1: Pseudocode of NACE

---

- **Actual World:** perceived_array = perceive_partial(world)
- **Observer:**

  s$_t$ = update_bird_view(s$_{t-1}$, perceived_array)

  $calculate(M_{change}, M_{mismatched}^{observation}, M_{mismatched}^{prediction})$
- **Hypothesizer:**

  Create new rules for which $w_+(r) = 1$.

  Update rule evidences according to $w_+(r)$ and $w_-(r)$.

  Choose rules $r_1$ with $w_+(r_1) > w_-(r_1)$ for which there does not exist a rule $r_2$ with same precondition and action, but different postcondition with $T_{exp}(r_2) > T_{exp}(r_1)$.
- **Planner utilizing Predictor:**

  $a_1, ..., a_n = BFS\_with\_Predictor(V(s) > 0)$

  $a_1^*, ..., a_n^* = BFS\_with\_Predictor(min(S(s)) < 1)$

  //whereby BFS_with_Predictor is bounded breadth first search with Predictor as state transition function

  If found($a_1, ..., a_n$):, return $a_1, ..., a_n$

  If found($a_1^*, ..., a_n^*$):, return $a_1^*, ..., a_n^*$

  Else, perform a random action

---

## APPENDIX E    HYPERPARAMETER DETAILS

### E.1    FOUNDATIONAL ALGORITHMS

#### E.1.1    CORE MODELS AND THEIR MECHANISMS

- **Deep Q-Network (DQN):** DQN integrates deep neural networks with classical Q-learning, making it effective for handling large state spaces. To stabilize training, DQN uses experience replay and a separate target network. The Q-value update in DQN follows:

$$Q(s, a) \leftarrow Q(s, a) + \alpha\big(R(s) + \gamma \max_{a'} Q(s', a') - Q(s, a)\big)$$

  where:
  - $s, a$: current state and action,
  - $s', a'$: next state and action,
  - $R(s)$: reward received,
  - $\gamma$: discount factor for future rewards,
  - $\alpha$: learning rate.

- **Advantage Actor-Critic (A2C):** A2C builds on the actor-critic framework, synchronizing multiple parallel learners to reduce variance in policy updates. It calculates an **advantage function** to evaluate actions relative to the current policy's value estimate, stabilizing training but requiring frequent environmental interactions due to its on-policy nature.

  **Advantage Function:**
$$A(s, a) = Q(s, a) - V(s)$$

  **Policy Update:** The policy is updated using the gradient:
$$\theta \leftarrow \theta + \alpha \nabla_\theta \log \pi_\theta(a|s) A(s, a)$$

  where:
  - $Q(s, a)$: action-value function,
  - $V(s)$: state-value function,
  - $\pi_\theta(a|s)$: policy parameterized by $\theta$,
  - $\alpha$: learning rate.

- **Trust Region Policy Optimization (TRPO):** TRPO addresses stability in policy updates by enforcing a trust region constraint, ensuring small policy changes during optimization. This constraint is implemented via a KL-divergence bound, preventing drastic shifts in behavior but requiring computationally expensive second-order optimization.

  **Objective Function:**

  $$\max_\theta \mathbb{E}_{s \sim \pi_{\theta_{\text{old}}}} \left[ \frac{\pi_\theta(a|s)}{\pi_{\theta_{\text{old}}}(a|s)} A(s,a) \right]$$

  **Constraint:**

  $$\mathbb{E}_{s \sim \pi_{\theta_{\text{old}}}} \left[ D_{\text{KL}}(\pi_{\theta_{\text{old}}} || \pi_\theta) \right] \leq \delta$$

  where:

  - $\pi_\theta(a|s)$: new policy,
  - $\pi_{\theta_{\text{old}}}(a|s)$: previous policy,
  - $A(s,a)$: advantage function,
  - $D_{\text{KL}}$: KL-divergence,
  - $\delta$: trust region size.

- **Proximal Policy Optimization (PPO):** PPO refines TRPO by introducing a clipped surrogate objective, which simplifies computation and allows for multiple updates per batch. This approach improves data utilization while maintaining policy stability.

  **Clipped Surrogate Objective:**

  $$\max_\theta \mathbb{E}_{s,a} \left[ \min \left( \frac{\pi_\theta(a|s)}{\pi_{\theta_{\text{old}}}(a|s)} A(s,a), \text{clip}\left( \frac{\pi_\theta(a|s)}{\pi_{\theta_{\text{old}}}(a|s)}, 1-\epsilon, 1+\epsilon \right) A(s,a) \right) \right]$$

  where:

  - $\pi_\theta(a|s)$: new policy,
  - $\pi_{\theta_{\text{old}}}(a|s)$: old policy,
  - $A(s,a)$: advantage function,
  - $\epsilon$: clipping threshold.

### E.1.2 HYPERPARAMETER CONFIGURATION FOR FOUNDATIONAL ALGORITHMS

We utilize the Stable Baselines3 framework (Raffin et al., 2021) to train and evaluate foundational algorithms, leveraging its pre-implemented models and customizable configurations. All algorithms use the same convolutional neural network architecture to process observations, ensuring consistency across experiments. The hyperparameters for each algorithm were selected based on achieving the best average performance across all tasks, rather than optimizing for a single task, to ensure generalizability. The details of the network architecture and training setup for each algorithm are outlined below.

**Network Architecture:** Observations ($7 \times 7 \times 3$) from the Minigrid environment are processed through four convolutional layers. Each layer is configured as follows:

- Kernel size: $2 \times 2$

- Activation: ReLU

- Increasing number of filters: 16, 32, 64, and 128

The output of the final convolutional layer is flattened and passed to a fully connected layer with:

- Output dimension: 128

- Activation: ReLU

**Training Configurations:**

| Parameter | DQN |
|---|---|
| Learning rate | 0.0001 |
| Buffer size | 1,000,000 |
| Learning starts | 100 |
| Batch size | 32 |
| Soft update coefficient | 1 |
| Discount factor | 0.99 |
| Train frequency | 4 |
| Gradient steps | 1 |
| Target update interval | 10,000 |
| Exploration fraction | 0.1 |
| Initial exploration epsilon | 1.0 |
| Final exploration epsilon | 0.05 |
| Max gradient norm | 10.0 |

| Parameter | A2C |
|---|---|
| Learning rate | 0.0007 |
| Number of steps | 5 |
| Discount factor | 0.99 |
| Entropy coefficient | 0.0 |
| Value function coefficient | 0.5 |
| Max gradient norm | 0.5 |

Table 3: DQN and A2C Training Parameters

| Parameter | TRPO |
|---|---|
| Learning rate | 0.001 |
| Number of steps | 2048 |
| Batch size | 128 |
| Discount factor | 0.99 |
| Conjugate gradient max steps | 15 |
| Conjugate gradient damping | 0.1 |
| Line search shrinking factor | 0.8 |
| Line search max iterations | 10 |
| Number of critic updates | 10 |
| Target KL divergence | 0.01 |

| Parameter | PPO |
|---|---|
| Learning rate | 0.0003 |
| Number of steps | 2048 |
| Batch size | 64 |
| Number of epochs | 10 |
| Discount factor | 0.99 |
| Clip range | 0.2 |
| Entropy coefficient | 0.0 |
| Value function coefficient | 0.5 |
| Max gradient norm | 0.5 |

Table 4: TRPO and PPO Training Parameters

## E.2 MODEL-BASED ALGORITHM: DREAMERV3

DreamerV3 is a model-based RL algorithm designed to enhance sample efficiency by learning a latent world model of the environment. It optimizes both the world model and the policy within the latent space, reducing the computational demands of interacting with the environment.

**World Model:** The latent dynamics model predicts future latent states $z$ based on prior latent state $z_{t-1}$, action $a_{t-1}$, and reward $R_{t-1}$. This model facilitates long-term planning without requiring explicit rollouts in the actual environment.

**Policy Optimization:** The policy maximizes expected rewards in the learned latent space by leveraging the dynamics model to simulate trajectories. Policy updates use gradient-based methods informed by imagined rollouts.

**Loss Function:**

$$\mathcal{L}_{\text{DreamerV3}} = \mathcal{L}_{\text{Reconstruction}} + \mathcal{L}_{\text{Dynamics}} + \mathcal{L}_{\text{Policy}}$$

where:

- $\mathcal{L}_{\text{Reconstruction}}$: Measures the accuracy of reconstructing environment observations,
- $\mathcal{L}_{\text{Dynamics}}$: Captures consistency in latent state transitions,
- $\mathcal{L}_{\text{Policy}}$: Maximizes imagined rewards.

**Hyperparameter Configuration for DreamerV3:** The hyperparameter configuration has been chosen to match the settings provided in `https://github.com/qxcv/dreamerv3`. To avoid redundancy and maintain brevity, we do not include the full configuration here due to its extensive nature.

### E.3 MODEL-FREE EXPLORATION AND SCALABILITY EXTENSIONS

All experiments for the other model-free methods are based on the Torchbeast implementation of IMPALA (Espeholt et al., 2018), which has been modified to support intrinsic reward algorithms as described in Raileanu & Rocktäschel (2020) and Campero et al. (2021). The hyperparameters were selected following the configurations used in these references. For clarity, we first list the values shared across all algorithms, followed by the specific details unique to each one.

#### E.3.1 SHARED HYPERPARAMETERS

- **Network Architecture:** Observations ($7 \times 7 \times 3$ for Minigrid) are processed through three convolutional layers:
    - Number of filters: 32 per layer
    - Kernel size: $3 \times 3$
    - Stride: 2
    - Padding: 1
    - Activation: Exponential Linear Unit (ELU)

  The output of the convolutional layers is passed to:
    - An LSTM layer to address partial observability by maintaining temporal dependencies and encoding sequences of observations.
    - A fully connected layer for computing:
        * Policy logits: Unnormalized scores for each action, converted to probabilities using a softmax function.
        * Value estimates: Predictions of expected future returns, used in actor-critic methods.

- **Training Setup:**
    - Number of actors: 40
    - Number of buffers: 80
    - Unroll length: 100
    - Number of learner threads: 4
    - Batch size: 32
    - Discount factor: 0.99
    - Learning rate: 0.0001
    - Policy entropy loss: 0.0005
    - Gradient clipping: Norm of 40
    - Save interval: Every 20 minutes

- **Special Parameters (Only When Applicable):**
    - Count reset probability: 0.001 (COUNT, RIDE)
    - Hash bits: 128 (COUNT)

#### E.3.2 INTRINSIC REWARDS AND COEFFICIENTS

Intrinsic rewards address sparse rewards and inefficient exploration. Each algorithm applies scaling coefficients to normalize its intrinsic rewards. Additionally, all techniques incorporate policies enhanced with LSTMs to address partial observability by maintaining memory of past observations and actions.

- **IMPALA:** No intrinsic reward ($r_i = 0.0$).
- **COUNT:** $r_i = 0.005$.
- **RIDE:** $r_i = 0.1$.
- **CURIOSITY:** $r_i = 0.1$.
- **RND:** $r_i = 0.1$.
- **AMIGO:** $r_i = 0.1$ (applies to the teacher's intrinsic rewards).

The formal definitions of the intrinsic rewards are:

**COUNT:** The intrinsic reward is based on state visitation counts, encouraging exploration of less-visited states:

$$r_i = \frac{1}{N(s_0)},$$

where $N(s_0)$ is the (pseudo)count of visits to state $s_0$. Counts are never reset during training.

**RIDE (Rewarding Impact-Driven Exploration):** The intrinsic reward combines state novelty and state-change impact:

$$r_i = \|\phi(s) - \phi(s_0)\|_2 \cdot \frac{1}{N(s_0)},$$

where $\phi$ is trained to minimize both forward and inverse dynamics prediction errors. Counts $N(s_0)$ are reset at the beginning of each episode.

**CURIOSITY:** The intrinsic reward comes from the prediction error of a forward dynamics model $f$, which predicts the next state embedding $\phi(s_0)$ from the current embedding $\phi(s)$ and action $a$:

$$r_i = \|f(\phi(s), a) - \phi(s_0)\|_2.$$

**RND (Random Network Distillation):** The intrinsic reward is computed as the prediction error of a trainable network $\phi$ attempting to match the output of a fixed random network $\hat{\phi}$:

$$r_i = \|\phi(s_0) - \hat{\phi}(s_0)\|_2.$$

**AMIGO:** The teacher policy generates goals $g$ for the agent, with rewards given as:

$$r_i = v(s_t, g) = \begin{cases} +1 & \text{if } s_t \text{ satisfies } g, \\ 0 & \text{otherwise.} \end{cases}$$

The total reward is a weighted sum of intrinsic and extrinsic rewards:

$$r_t = \beta r_i + \alpha r_e, \quad \text{with } \beta = 0.3, \alpha = 0.7.$$

**Algorithm-Specific Hyperparameters and Architectures:**

| Algorithm | Details |
|---|---|
| IMPALA (Baseline) | Intrinsic reward: None. Loss: Policy gradient, baseline, entropy. |
| COUNT | Intrinsic reward: State visitation counts. Count reset probability: $p = 0.001$. |
| CURIOSITY | Intrinsic reward: Forward prediction error. State embedding: 256-dimensional. The Forward model predicts the next state. Inverse model predicts actions. Loss weights: Forward 10.0, Inverse 0.1. |
| RIDE | Intrinsic reward: Counts $\times$ norm of state change. Modules: Same as CURIOSITY. |
| RND | Intrinsic reward: Prediction error. Target net: Fixed embeddings. Predictor: Trained for target match. Loss weight: 0.1. |
| AMIGO | Intrinsic reward: Teacher-generated. Reward coefficients: Intrinsic $\beta = 0.3$, Extrinsic $\alpha = 0.7$. Batch size: 150. Entropy cost: 0.05. Threshold: $-0.5$. |

## APPENDIX F  HARDWARE AND RUNTIME

In this section, we describe the hardware setup used to run the techniques and provide runtime characteristics, including the duration of 5000 representative timesteps for each technique. While we report this information for reproducibility, we emphasize that the focus of our analysis is not on computational cost, but rather on sample efficiency.

- CPU: Intel Core i7-9750H with 32GB RAM
- GPU: Geforce GTX-1660 Ti with 6GB RAM

| Algorithm | Empty-16x16 | DistShift2 | LavaGapS7 | SimpleCrossingS11N5 | Unlock | DoorKey-8x8 | Average |
|---|---|---|---|---|---|---|---|
| TRPO | 751.46 | 84.89 | 84.05 | 374.76 | 241.70 | 489.37 | 337.04 |
| PPO | 745.70 | 149.06 | 128.53 | 384.06 | 240.66 | 493.73 | 357.29 |
| A2C | 625.22 | 181.67 | 157.37 | 323.45 | 207.82 | 413.42 | 318.83 |
| DQN | 593.23 | 130.77 | 122.55 | 284.60 | 156.81 | 321.42 | 268.90 |
| IMPALA | 268.98 | 282.99 | 274.11 | 262.70 | 267.95 | 249.48 | 267.70 |
| COUNT | 357.52 | 403.85 | 281.94 | 332.05 | 257.58 | 273.81 | 317.13 |
| RIDE | 361.53 | 354.95 | 288.58 | 377.65 | 338.89 | 265.41 | 331.17 |
| CURIOSITY | 392.20 | 299.62 | 329.57 | 358.67 | 323.51 | 296.38 | 333.66 |
| RND | 357.41 | 298.23 | 308.14 | 378.99 | 317.51 | 373.73 | 339.67 |
| AMIGo | 49.32 | 77.84 | 68.38 | 47.68 | 59.88 | 44.30 | 57.90 |
| DreamerV3 | 102.38 | 78.67 | 84.39 | 76.01 | 73.38 | 78.62 | 82.91 |
| NACE | 314.12 | 301.93 | 302.06 | 301.58 | 302.24 | 360.11 | 313.67 |

Table 5: Runtime (in seconds) for different algorithms across MiniGrid v0 environments.

## APPENDIX G  EXAMPLE ENVIRONMENT WITH LEARNED RULES

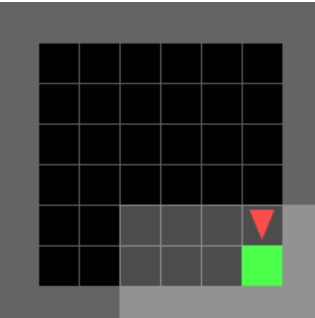

Figure 9: Illustration of Minigrid-Empty-8x8

The following are all the rules NACE learns in the Minigrid-Empty-8x8 environment as illustrated in Figure 9:

```
Agent interacting with goal location:
<(v=[1], c[ 0, 0 ]='x', c[ 0, 1 ]='H', ^down ) =/> (v=[0], c[ 0, 0 ]='.', R(r)=1)>.
<(v=[1], c[-1, 0 ]='H', c[ 0, 0 ]='x', ^left ) =/> (v=[0], c[ 0, 0 ]='.', R(r)=1)>.
<(v=[1], c[ 0,-1 ]='H', c[ 0, 0 ]='x', ^up  ) =/> (v=[0], c[ 0, 0 ]='.', R(r)=1)>.
<(v=[1], c[ 0, 0 ]='x', c[ 1, 0 ]='H', ^right) =/> (v=[0], c[ 0, 0 ]='.', R(r)=1)>.
Goal location interacting with agent:
<(v=[1], c[ 0,-1 ]='x', c[ 0, 0 ]='H', ^down ) =/> (v=[0], c[ 0, 0 ]='.', R(r)=1)>.
<(v=[1], c[ 0, 0 ]='H', c[ 1, 0 ]='x', ^left ) =/> (v=[0], c[ 0, 0 ]='.', R(r)=1)>.
<(v=[1], c[ 0, 0 ]='H', c[ 0, 1 ]='x', ^up  ) =/> (v=[0], c[ 0, 0 ]='.', R(r)=1)>.
<(v=[1], c[-1, 0 ]='x', c[ 0, 0 ]='H', ^right) =/> (v=[0], c[ 0, 0 ]='.', R(r)=1)>.
Agent interacting with empty space:
<(v=[1], c[ 0, 0 ]=' ', c[ 0, 1 ]='x', ^up  ) =/> (v=[1], c[ 0, 0 ]='x', R(r)=0)>.
<(v=[1], c[-1, 0 ]='x', c[ 0, 0 ]=' ', ^right) =/> (v=[1], c[ 0, 0 ]='x', R(r)=0)>.
<(v=[1], c[ 0,-1 ]='x', c[ 0, 0 ]=' ', ^down ) =/> (v=[1], c[ 0, 0 ]='x', R(r)=0)>.
<(v=[1], c[ 0, 0 ]=' ', c[ 1, 0 ]='x', ^left ) =/> (v=[1], c[ 0, 0 ]='x', R(r)=0)>.
Empty space interacting with agent:
<(v=[1], c[ 0,-1 ]=' ', c[ 0, 0 ]='x', ^up  ) =/> (v=[1], c[ 0, 0 ]=' ', R(r)=0)>.
<(v=[1], c[ 0, 0 ]='x', c[ 1, 0 ]=' ', ^right) =/> (v=[1], c[ 0, 0 ]=' ', R(r)=0)>.
<(v=[1], c[ 0, 0 ]='x', c[ 0, 1 ]=' ', ^down ) =/> (v=[1], c[ 0, 0 ]=' ', R(r)=0)>.
```

```
<(v=[1], c[-1, 0 ]=' ', c[ 0, 0 ]='x', ^left ) =/> (v=[1], c[ 0, 0 ]=' ', R(r)=0)>.
Agent interacting with wall:
<(v=[1], c[ 0,-1 ]='o', c[ 0, 0 ]='x', ^up   ) =/> (v=[1], c[ 0, 0 ]='x', R(r)=0)>.
<(v=[1], c[ 0, 0 ]='x', c[ 1, 0 ]='o', ^right) =/> (v=[1], c[ 0, 0 ]='x', R(r)=0)>.
<(v=[1], c[ 0, 0 ]='x', c[ 0, 1 ]='o', ^down ) =/> (v=[1], c[ 0, 0 ]='x', R(r)=0)>.
<(v=[1], c[-1, 0 ]='o', c[ 0, 0 ]='x', ^left ) =/> (v=[1], c[ 0, 0 ]='x', R(r)=0)>.
Wall interacting with agent:
<(v=[1], c[ 0, 0 ]='o', c[ 1, 0 ]='x', ^left ) =/> (v=[1], c[ 0, 0 ]='o', R(r)=0)>.
<(v=[1], c[ 0, 0 ]='o', c[ 0, 1 ]='x', ^up   ) =/> (v=[1], c[ 0, 0 ]='o', R(r)=0)>.
<(v=[1], c[-1, 0 ]='x', c[ 0, 0 ]='o', ^right) =/> (v=[1], c[ 0, 0 ]='o', R(r)=0)>.
<(v=[1], c[ 0,-1 ]='x', c[ 0, 0 ]='o', ^down ) =/> (v=[1], c[ 0, 0 ]='o', R(r)=0)>.
```

The number of learned rules required to deal with the Minigrid environments typically varies between 16 (minimum with walls and free space) and usually less than 100, depending on the number of cell types, whereby for two cell types to interact with $m$ actions, at least $2 * m$ additional rules are learned.

## APPENDIX H    TEST ENVIRONMENTS

Prior to moving to Minigrid, NACE was first tested with internal levels:

- Level 1: Food collection. In this level, as depicted in Figure 10, the agent needs to collect food while avoiding walls.

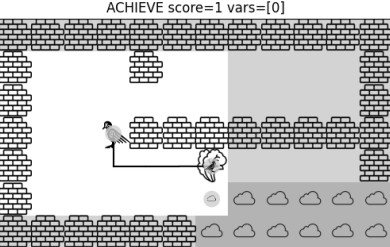

Figure 10: Food collection with walls

- Level 2: Doors and keys. In this level, as depicted in Figure 11, the agent needs to open doors with keys in order to collect batteries.

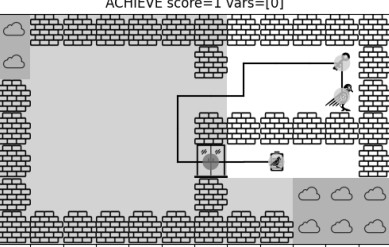

Figure 11: Battery collection level

- Level 3: A pong game in a grid world as illustrated in Figure 12, where the agent can only move vertically and needs to catch the ball by predicting its movement.

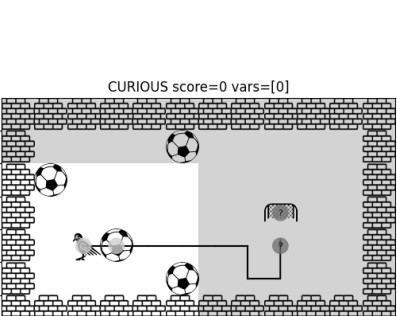

Figure 12: Pong game

- Level 4: Egg delivery. In this level, as depicted in Figure 13, the agent needs to deliver eggs to the chicken.

Figure 13: Egg collection level

- Level 5: Soccer level. In this level, as depicted in Figure 14, the agent needs to learn to shoot balls into the goal.

Figure 14: Soccer level

- Level 6: Food collection while avoiding electric fences. In this level, as depicted in Figure 15, the agent needs to collect food while avoiding electric fences.

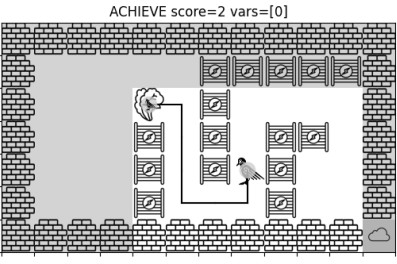

Figure 15: Food collection electric fence

- Level 7: Sokoban-like puzzle world (Dor & Zwick, 1999). In this level, as depicted in Figure 16, the agent needs to utilize the interaction properties of many different object types to successfully collect batteries:

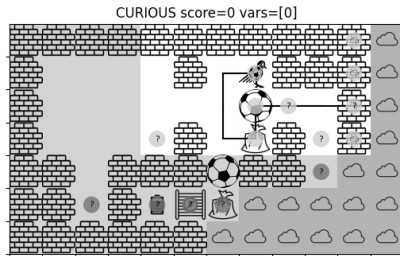

Figure 16: Sokoban-like level