# OpenReview forum: "Rule-Based Grid World Exploration under Uncertainty"
_ICLR.cc/2026/Conference — Submitted to ICLR 2026_

### Official Review · Reviewer_7QQo · 2025-10-17

**Soundness:** 1
**Presentation:** 1
**Contribution:** 1
**Rating:** 0
**Confidence:** 3

**Summary:**

The work proposes NACE, an RL algorithm deeply specialized in gridworld scenarios where the learning process is carried out by generating rules of the form (precondition, action) -> consequence. The authors evaluate the method in a collection of gridworlds and claim the method beats standard baseline DRL methods.

**Strengths:**

Paper is well-written and the authors considered a wide range of DRL methods to compare against.

**Weaknesses:**

- The paper simply completely ignores a large body of works in Relational MDPs and even Object-Oriented MDPs  that have a fairly similar learning process and motivation, some of the works dating all the way back to the last century:

Džeroski, Sašo, Luc De Raedt, and Kurt Driessens. "Relational reinforcement learning." Machine learning 43.1 (2001): 7-52.
Diuk, Carlos, Andre Cohen, and Michael L. Littman. "An object-oriented representation for efficient reinforcement learning." Proceedings of the 25th international conference on Machine learning. 2008.
Da Silva, Felipe Leno, Ruben Glatt, and Anna Helena Reali Costa. "MOO-MDP: An object-oriented representation for cooperative multiagent reinforcement learning." IEEE Transactions on Cybernetics 49.2 (2017): 567-579.

The reason why this line of work faded a bit out of fashion is because any such method needs extensive work in modeling and providing useful descriptions of the environment, which might be very challenging to do in an environment more complicated than a gridworld.

I would say that a meaningful work following this direction should 1) think of a way of incorporating this type of learning to the learning process in a way that it works for way more challenging domains than gridworlds, and demonstrate that the agent solves environments that cannot be solved with tabular representation; 2) compare it against meaningful RMDP/OO-MDPs baselines, comparing against "normal" DRL algorithms is not fair because NACE makes use of a very handcrafted and specialized modeling of the environment so that it works well, whereas the baselines are design to work with minimal work in the state representation.

**Questions:**

No specific question

---

### Official Review · Reviewer_tNBu · 2025-10-22

**Soundness:** 2
**Presentation:** 1
**Contribution:** 3
**Rating:** 2
**Confidence:** 3

**Summary:**

A new model-based RL method for grid environments. The main contribution is in the introduction of a novel model learning component, which is based in rule learning (rather than the popular alternative of DNNs).

**Strengths:**

Strong performance on domains that contain important challenges for RL (sparse-reward and in-distribution generalization).

Interesting, more structured alternative for model-learning which seems like a promising direction.

**Weaknesses:**

**Evaluation:**

The method is evaluated in three types of environments:
1. Sparse reward environments.
2. Sparse reward + (in-distribution?) generalization environments.
3. Harder Sparse reward + generalization environments.

However, many of the baselines are not equipped with deep-exploration mechanisms (A2C, DQN, PPO, TRPO...), making their inclusion irrelevant.
If the intent is to show that the method is fundementally more efficient than deep-RL methods, I would like additionally:
1. Evaluation in rich-reward (/+complex dynamics) environments.
2. Comparison to stronger deep-RL baselines (for example, MuZero) rather than older and non-SOTA (DQN, A2C).

In addition, relevant strong deep exploration baselines are not included:
1. Bootstrapped DQN [1].
2. Planning to explore in model-based algorithms [2,3].
3. Pure counting based methods that are applicable to tabular / grid environments. If the number of states is very large, hash-based counting [4] is applicable and very strong.

Finally, there are other methods for non-deep-RL for tabular / grid environments, such as PSRL [5], that are extremely powerful. Is there a reason to use the method proposed in this paper over these methods?

Together, it makes it hard for me to judge what is the significance of the contribution made in the paper.

**Clarity:**
1. There is no background / preliminaries section, that introduces the rule-based framework in which this method is based. I think such a background will significantly help the reader, especially for RL-readers from other areas. It will also contribute to the reader's ability to understand the exact novelty and contributions of the paper.
2. I do not understand the planner as it is explained (example questions in the questions section). If it is based in existing planners, they are not referenced / cited.
3. Is the goal observable at every env., at every state? (If not, the task becomes searching through all known goal locations, and POMDP methods should be added to the evaluation.)
4. What is the (ballpark) size of the statespace of the environments evaluated on?
5. The 11 baselines that are compared against are very briefly introduced and with very little detail, making it hard to judge what is the strength of the results.
6. Additional limitatons:
    1. Is the method limited to deterministic environments?
    2. Is the method limited by the size of the state space? (ie only applicable for small grid environments?)
    3. Is the method limited to positive rewards?

Together, this makes it difficult for me to evaluate whether the method is sound.

[1] Osband, Ian, et al. "Deep exploration via bootstrapped DQN." NuerIPS (2016).

[2] Sekar, Ramanan, et al. "Planning to explore via self-supervised world models." ICML 2020.

[3] Oren, Yaniv, et al. "Epistemic Monte Carlo Tree Search." ICLR 2025.

[4] Tang, Haoran, et al. "# exploration: A study of count-based exploration for deep reinforcement learning." NeurIPS 2017.

[5] Osband, Ian, Daniel Russo, and Benjamin Van Roy. "(More) efficient reinforcement learning via posterior sampling."NeurIPS 2013.

**Questions:**

1. Is it necessary to use the planner proposed by the method, or can other planners (such as MCTS [1]) be used here? If other planners are applicable, why was this planner used?
2. What does this mean mathematically? (line 265->) "a combined search objective consisting of two components: it searches for states
resulting from the different action sequences for futures that lead to the max. Expected return or, if not existing, the lowest state match value." In addition, "max. Expected return" should be "maximum expected return" (full word, no capitalization of the E). And in fact, had value been defined beforehand in a preliminaries section, it could have been directly used here for increased clarity.
3. It's not clear to me what the authors mean here (line 268->): "Hence, it applies a key RL principle to maximize the expected long-term return (Sutton et al., 1999), with the policy determined by the considered action sequence: $\pi(t)$ = at for t = 1, 2, . . . , n whereby n is smalleror-equal (dependent on where the optimum is found) to the maximum planning horizon". I assume it should be $\pi(s_t)$? The notation has not been defined which makes this harder to follow.
4. It's not clear to me what the authors mean here (line 278->): "According to this definition, if no return greater than zero can be obtained for any considered action sequence, the system instead plans for a future state of lowest state match value, while maintaining the “uncertainty-at-the-last-step” condition:"

See the weaknesses section for additional questions.

A minor comment is that throughout the paper there are many \cite (references to the authors) that shoudl be \citep (references to the work). For example, in the paragraph starting at line 091.

[1] Świechowski, Maciej, et al. "Monte Carlo tree search: A review of recent modifications and applications." Artificial Intelligence Review 56.3 (2023): 2497-2562.

---

### Official Review · Reviewer_EC9e · 2025-10-30

**Soundness:** 4
**Presentation:** 2
**Contribution:** 3
**Rating:** 6
**Confidence:** 4

**Summary:**

The paper "Rule-Based Grid World Exploration under Uncertainty" introduces a reinforcement learning agent for grid world environments that learns precondition & consequence pairs to inform decisions. The authors create the framework "NACE" for observing and scoring changes caused by the agent's actions to formulate relevant rules that help predict expected returns and drive exploration. To limit the number of rules created, certain invariants and biases are introduced with extensive reasoning. In thorough experiments they compare NACE against twelve deep reinforcement learning agents (including state-of-the-art like DramerV3) in some of the established MiniGrid benchmarks. They achieve impressive sample efficiency increases of up to three magnitudes compared to the second most efficient model to reach similar performance across all benchmarks.

**Strengths:**

Your paper introduces an original framework to learn rules supplemented with the necessary theory. The thorough literature review combined with detailed and understandable reasoning why biases were selected clearly show the motivation behind NACE. On top of that, the modular construction of your system (Observer, Hypothesizer, Planner, Predictor) helps to understand the general approach and allows for further research in each step of the algorithm. Finally, your evaluation shows the impressive sample efficiency of rule-based approaches, which make this a relevant research area.

**Weaknesses:**

The obvious weakness of the presented approach (as mentioned in the paper) is that it does not easily generalize to other environments. Since "pathways for scaling to complex environments" are illuded to in the introduction, the lack of further discussion in the paper (except for neural implementations mentioned in the conclusion and removing actions from preconditions in Appendix A) is unfortunate.

Furthermore, the lack of example rules constructed by the agent in the main paper makes their advantages hard to grasp for readers that are new to rule-based systems. The rules provided in Appendix G are also presented without describing the meaning of symbols (like H,x,o). For some of the definitions in chapters 3.3 and 3.4  like $C(c), S(s)$ and $T_{exp}$ no clear reasoning on their design is provided or it is unclear how the reasoning is reflected in the formulas, which makes it more difficult to follow the logic behind them.

**Questions:**

It seems like the interpretability aspect of this work is barely touched upon in the paper. However, it appears to be a significant advantage of the proposed methods, since the agent's intention can easily be extracted from the learned rules.

From my understanding the definition for the policy $\pi(t)$ around line 270 is very imprecise. I am particularly confused by the condition $V^\pi(s_0)=R_{exp}$ for the first case, since that is simply the definition for the value function. From the consequent paragraph it appears like the condition should actually be "if $V^\pi(s_0)>0$" and $R_\text{exp}$ should simply be replaced by $V^\pi(s_0)$ in line 273. On top of that, the "$<1$" in line 271 is unnecessary, since it looks like $S(s_n) <1$ is a boolean statement that is being optimized.

The notation for the spatial state appears to be conflicting throughout chapter 3. It is introduced as a matrix with $m\times n$ entries corresponding to the grid, but in line 194 you only refer to cells $1,\dots,m$ for each state. Specifically in this section, I was confused by the wording "[...] is the maximum value of its cells" since my understanding was that each state contains all cells.

In your description of the Planner around line 285, it would help clarity if the complexity was expressed in previously mentioned terms like the number of rules, instead of the abstract description for any graph, especially since $V$ is already used for the value function.

(In general, the notation sometimes appears convoluted with indeces and the mixture of letter & word notation, math & text mode in some equations contributes to decreased readability.)

---

### Official Review · Reviewer_gjdX · 2025-11-01

**Soundness:** 2
**Presentation:** 3
**Contribution:** 3
**Rating:** 4
**Confidence:** 4

**Summary:**

The paper develops a framework (NACE) to learn local transition rules in partially observable grid worlds and use constrained breath-first planning to explore and solve the grid worlds in question. The approach is evaluated in 5 MiniGrid domains against an impressive range of RL approaches, and shows the approach is around 3 magnitudes more sample efficient.

**Strengths:**

The paper is unusual for an ICLR submission: as far as I can see the proposed methods does not use any neural networks. Instead a list of local rules for each gridworld cell is learned and evaluated. I still like the main idea, despite some of the flaws (see below). The idea of using states in which no clear rewarded solution can be found for rule exploration is very nice. The results look very good on the surface, and show a consistent improvement of sample efficiency of ~1000 times over DreamerV3, and even more over model-free DRL methods.

**Weaknesses:**

At the core of NACE is a breath-first search algorithm with the learned world model. This means the decision policy should require significantly *more* time during deployment than the neural-network policies learned by the baselines. This begs the question: do the authors compare the correct methods here? I could not find any indication that NACE produces neural-network policies (although that would be a great idea to try), and so I am forced to conclude that the impressive 3 orders of magnitude are a byproduct of comparing apples with oranges.

I might be wrong here, but I could not find any mention how new rules are **created** by the Hypothesizer. This is crucial. Without creating more rules the agent needs to start with a superset of the true world rules, and the curiosity model would not make any sense, so I assume that the authors did this somehow (they mention that fact multiple times). But generating decent candidate rules appears to be the hardest part of NACE, so I cannot recommend acceptance of the paper without a detailed description of it. The authors also missed references [1,2] to relation RL literature, that learn (stochastic) rules similar to NACE, including how to use them for exploration [3].

Using counts of consistent/inconsistent observed rule applications seems to be a dangerous design decision. As the agent changes its policy, the frequency of observed transition changes. An agent who only arrives at a door after picking up the key might after a while learn the rule that the door always opens. When this changes the agent's behavior, the obvious ineffectiveness of that rule would quickly lower its truth expectation $T_{exp}$, but this could lead to cyclic performance crashes. The presence of another rule that has possession of the key as a precondition would override this problem, though, as it is always right and therefore has a higher truth expectation. However, the same can be said for any number of overfitting rules that are only applicable in the specific situations observed during training. This does not show up in the evaluation, as every episode has a random context, but training only on a small subset of contexts (e.g. topologies or goal positions) can easily lead to overfitting rules.

Again, I could be wrong, but as far as I can say, the proposed planner just performs depth- and width-bounded breadth-first search, without any neural network heuristics. This is a huge problem, as some states might be too far away from any goal state to be rewarded. In these cases NACE could therefore *never* do anything but act randomly (which is not even secured against catastrophic actions like jumping into lava). Here a simple extension like AlphaZero's value function for the leaf nodes would improve the approach immensely. The authors should consider AlphaZero as a planner in general, as it is much more time-efficient and allows to make anytime decisions.

The presented approach only makes sense in strictly deterministic environments, otherwise the "update of unobserved grid cells" falls apart. This must be discussed! Furthermore, it ignores uncertainty based on partial observability. For example, what if there is an "enemy" that moves deterministically through the gridworld. If the agent has not seen it, the planner would assume there is no enemy in the world. This can lead to suboptimal decisions, e.g. choosing a narrow corridor where the enemy cannot be evaded, because it is shorter. Using AlphaZero as planner should fix this, as the value function can learn that on average the narrow tunnel has a lower expected return.

**Detailed comments**
- Define a general POMDP in the beginning of Section 3, followed by a clear definition how your gridworlds map onto that.
- Define how and when your rules are created!
- The *state match* measure only considers preconditions. While this is good for (I assume) creation of new rules, it does not consider cases where two rules match a state but have different consequences. Maybe include this as an additional measure?
- In the definition of $S(s)$, the max should not be italic and $m$ should probably be another variable, like the number of observable cells.
- How do you learn the reward component of a rule? If many rules produce reward at the same time it should be very tough to disentangle who was responsible for what.
- l.266: "max. Expected" -> "max expected"
- The definition of $\pi(t)$ in linees 270-272 (please enumerate your equations) is ambiguous. What does it mean that $V^\pi(s_0) = R_{exp}$? You mean that $V^\pi(s_0) > 0$, i.e., that a leaf found a rewarded goal-state, but this cannot be deduced from the equation, and $V^\pi$ is never defined.
- The "uncertainty-at-the-last-step" is simply an assumption that leafs with $S(s_n) < 1$ are terminal for the computation of $S$. Please rephrase this.
- The description of the computational complexity of NACE is misleading: please clarify that breadth-first search scales exponentially in the tree depth. I could not determine how your width-bounded search works, as in many situations none of the leafs are rewarded when the width-bound in reached, so without a neural network value it is not clear to me which nodes are kept and which are dropped.
- The term "inductive-bias", while technically correct, was confusing to me, as I associated inductive biases in neural network training (and kept waiting for the neural networks to show up). Please clarify this in the introduction.

**References**

[1] Pasula, H. M.; Zettlemoyer, L. S., and Kaelbling, L. P. Learning symbolic models of stochastic domains. Journal of Artificial Intelligence Research, 29:309–352, 2007.

[2] Lang, T. and Toussaint, M. Planning with noisy probabilistic relational rules. Journal of Artificial Intelligence Research, 39:1–49, 2010.

[3] Lang, T.; Toussaint, M., and Kersting, K. Exploration in relational domains for model-based reinforcement learning. Journal of Machine Learning Research, 13(1):3725–3768, 2012.

**Questions:**

- How does NACE create new rules?
- How do the curiosity module avoid being attracted to states where there is no observable change (e.g.running against a wall)? In those cases no rules should be active, so $S(s)$ should be small, even though there is nothing interesting in this state.

---

### Meta-Review · Area_Chair_CYu8 · 2026-01-04

**Summary:**

The paper introduces a model-based RL algorithm that learns a classical rule-based model of the form (precondition, action) -> consequence. The method is developed to operate in partially observable gridworlds. Reviewers have raised many concerns, raised from the fairness of the comparisons performed to specific design decisions in the algorithm itself, including its feasibility in stochastic environments. Importantly, the proposed approach is limited to gridworlds, as acknowledged in the paper. None of these issues were discussed, nor were the questions answered by the authors. Given the overwhelmingly negative reviews of this paper, I am recommending its rejection.

**Reviewer Concerns:**

No concerns were addressed.

**Reviewer Scores:**

The reviewers couldn't have changed their score.

---

### Decision · Program_Chairs · 2026-01-26

Reject